# Healthy Aging at Family Mealtimes: Associations of Clean Cooking, Protein Intake, and Dining Together with Mental Health of Chinese Older Adults amid COVID-19 Pandemic

**DOI:** 10.3390/ijerph20031672

**Published:** 2023-01-17

**Authors:** Shuai Zhou, Xiangying Ding, Janet Tsin Yee Leung

**Affiliations:** Department of Applied Social Sciences, The Hong Kong Polytechnic University, Hong Kong SAR, China

**Keywords:** healthy aging, mental health, diet, protein intake, clean cooking, family dining, older adults, Chinese

## Abstract

The present study aims to examine whether multiple dietary factors affect the mental health of older adults amid the COVID-19 pandemic. It proposes an integrative dietary framework that highlights environmental, nutritional, and social aspects of diet for healthy aging. Based on a sample of 7858 Chinese older adults, the associations between diet and depressive symptoms, along with the rural–urban divide, were examined using zero-inflated negative binomial regression. Overall, protein intake (incidence-rate ratio [IRR] = 0.89, *p* < 0.001), frequency of family dining together (IRR = 0.98, *p* < 0.001), and using tap water for cooking (IRR = 0.92, *p* < 0.01) were associated with lower incidence rates of depressive symptoms among older adults. Among rural older adults, frequency of family dining together (IRR = 0.97, *p* < 0.001) and tap water use (IRR = 0.89, *p* < 0.001) were associated with fewer depressive symptoms. However, urban residents who had a higher frequency of family dining together (IRR = 0.98, *p* < 0.05) and protein intake (IRR = 0.81, *p* < 0.001) exhibited fewer depressive symptoms. The findings revealed multifaceted dietary pathways towards healthy aging, which call for policies and interventions that improve diet quality for community-dwelling older adults.

## 1. Introduction

Inherent to the global policy agenda on population aging is the urgent priority of optimizing the health and well-being of older adults. It is estimated that the total number of people suffering from mental disorders was above 300 million in 2015, with older age groups at the highest risk of being depressed [1]. The outbreak of the COVID-19 pandemic increased the severity of this situation because older people accounted for a disproportionate number of hospital admission and death cases. Many older adults experienced an aggravation of mental health problems, such as depression [2], and deteriorated emotional well-being [3] during the initial period of the pandemic. Depression has been identified as a risk factor for frailty [4], a prodrome for dementia [5], and a leading contributor to mortality [6]. To address the mental health burden, researchers have identified a wealth of protective and modifiable factors, such as individual coping strategies [7] and social support resources [8].

Notably, several strands of research have explored the roles of cooking and eating in mental well-being. Environmental research has advocated for the notion of clean cooking and argued that using clean water and energy for cooking is salutary for sustainable livelihoods and human health [9,10]. In addition, nutrition research has indicated that food intake is crucial for maintaining mental health [11]. Amidst the COVID-19 pandemic, a global food crisis emerged due to the shortage of food supply and income shrinkage, which has substantially affected individual eating behaviors [12]. In 2021, the World Health Organization released an action framework for a healthy diet that emphasizes the necessity for policymakers to improve the availability and quality of food in public settings, such as community centers [13]. However, diet pertains to what people eat, how they cook, and, more importantly, whom they dine with. Emerging studies from east Asia, for example, showed that eating alone may increase the risk of depression [14] and frailty [15] in old age. Although researchers have touched upon the health effects of diet from diverse perspectives, their efforts have long been made in disconnect. A lack of conceptual synthesis has rendered comprehensive dietary interventions unviable. Hence, ascertaining the roles of dietary factors in shaping later-life well-being during the pandemic is important and has profound implications for public health policy and services.

The current study aimed at examining Chinese older adults’ mental health during the COVID-19 pandemic from an integrative diet perspective. Based on a population-based survey conducted in 2020, we identified multiple dietary factors associated with older adults’ mental health. We also compared the differences between rural and urban older adults to investigate the regional disparities in the mental health effects of diet. The findings from this study could inform research and policy in other countries where health risks related to cooking and eating are of increasing concern.

## 2. Literature Review

### 2.1. Multifaceted Diet and Healthy Aging

In a narrow sense, diet is defined as the intake pattern of foods [16]. Much research on healthy diet primarily focuses on the optimal intake of macronutrients [17]. Nonetheless, diet is not solely concerned with the food itself. A growing number of researchers have indicated that the understanding of diet goes beyond the nutritional value by exploring other dimensions, such as the social dimension and safety [18]. This study extends the multidimensional view of diet by integrating the environmental, nutritional, and social aspects of diet.

Environmental research on the water–energy–food nexus and sustainable livelihoods has argued that food security coupled with water and energy security is beneficial for sustainable livelihoods [10,19]. Recent initiatives of clean cooking also suggest that strategies for promoting public health should maximize the uptake of clean energy and safe water for cooking [9,20]. Conceptually, cooking fuels could be clustered into the following two groups: (a) clean fuels, including gaseous fuels, electricity, and other clean fuels; and (b) polluting fuels, such as solid fuels [21]. Previous studies have found that unclean cooking fuels are associated with a significantly higher risk of depression among older adults in low- and middle-income countries [22], including China [23,24], due to increased exposure to indoor air pollution. In addition, water insecurity, defined as inadequate or uncertain access to clean water, was found to be associated with risks of depression [25]. Using polluted or unclean water could aggravate health problems [26,27], especially in developing areas. In contrast, access to clean water is significantly associated with better mental health among older adults [23,28].

Nutrition is characterized by the food components or macronutrients people consume [16]. According to the health lifestyle theory, diet is among the most common health behaviors that could affect individual well-being [29]. Unhealthy dietary practices can become prevalent because older people may lack adequate knowledge of a healthy diet or neglect the nutritional guidelines [30]. Evidence shows that older adults may be more vulnerable to dietary risks, such as malnutrition [31]. Particularly, protein intake has gained growing scholarly attention. Some research indicates that a low intake of specific nutrients, such as protein, is associated with an increased likelihood of being frail in later life [32,33]. The frequent consumption of animal protein, such as milk, could decrease the risk of depression [34] among older adults.

The family dining experience is also a driving force of well-being in later life. The theory of commensality highlights that sharing food at the table creates a social space where bonding, equality, and reciprocity can be reinforced [35]. Family members, such as one’s partner and children, can help to prepare meals and infuse joy and pleasure into older adults’ meal experience [36]. In this respect, frequently eating together is a typical family ritual that could promote social integration [37], while a lack of mealtime companionship may signal social disengagement [38]. A British study shows that one in three households experience eating meals alone, of which a majority were single-person families [39]. It is further reported that depression symptoms were more common among older adults who ate alone [14,40]. However, less is known about older adults’ family dining experience and its mental health consequences in the context of the pandemic.

### 2.2. Rural and Urban Disparities

Habitual diet may vary significantly across contexts. In rural areas, access to food that supports a health-conscious diet may be restricted due to the lower availability of food services and high prices [41]. Evidence of nutritional status suggests that rural older adults had a lower intake of protein, fiber, and fruits than those dwelling in urban areas [42]. Regarding cooking technologies, urban residents are more likely to use gaseous fuels, while rural families tend to rely on unclean fuels, such as biomass [21]. In rural areas, a China-based study revealed that the proportion of rural communities using non-tap water still accounted for about three-quarters in 2011 [43]. Moreover, it was found that the eating behaviors of residents in developed areas are characterized by individualization [35]. To our knowledge, no prior research has systematically examined rural–urban divides in diet and mental health. It would be more meaningful to explore whether and how diet characteristics are associated with mental health outcomes by accounting for the disparities between rural and urban older adults.

### 2.3. The Present Study

Based on the health lifestyle theory, an environmental perspective, and the theory of commensality, this study proposes an integrative conceptual framework to investigate the mental health effects of diet in later life (Figure 1). Home-based cooking and eating are viewed as multifaced family activities that comprise individual food consumption (nutritional aspect), specific cooking environments (environmental aspect), and family meal rituals (social aspect). Like nutrition research, food intake is considered an inherent part of the diet because it is essential for functioning and survival. A significant hallmark of this framework is the recognition of environmental and social aspects of diet, which are equally important but relatively underexplored in the literature. Regarding the associations between diet and mental health, three sets of testable hypotheses are proposed below. (H1) The household environment hypothesis: *The use of clean energy and water for cooking is associated with better mental health of older adults*. (H2) Nutrition hypothesis: *Protein intake is associated with better mental health of older adults*. (H3) Commensality hypothesis: *The frequency of family dining together is associated with better mental health of older adults*. In addition, we propose a hypothesis related to the rural–urban divide (H4): *The associations between diet and mental health among older adults are conditioned on the area of residence*.

## 3. Materials and Methods

### 3.1. Data

Data were obtained from the sixth wave of the China Family Panel Studies (CFPS), a nationally representative and high-quality household survey conducted in mainland China. Launched in 2010, the CFPS successfully conducted interviews with 14,960 households from 25 provinces at the baseline survey [44]. The survey adopted a multi-stage probability proportional to size (PPS) sampling method and implicit stratification to improve representativeness and efficiency [45]. Five follow-up surveys were carried out to recapture all family members in baseline households biennially, facilitated by the computer-assisted telephone interviews system [46]. To ensure data quality, rigorous statistical checks, audio checks, and telephone call checks were conducted after each follow-up wave [47]. The sixth wave that incorporated brief instruments regarding nutrition intake was conducted from July to December 2020, soon after the initial lockdown period. Due to stringent social distancing measures, the vast majority of interviews were conducted via telephone calls. We supplemented the data with household-related information, such as household size, family income, and cooking environments, from the prior CFPS survey collected in 2018 and 2019.

Our sample was restricted to Chinese older adults, who were aged 50 years or above and had valid household-related information in the fifth wave. Among the 10,265 eligible respondents, those with missing data on depressive symptoms (*n* = 1802), diet characteristics (*n* = 467), and control variables (*n* = 138) were further excluded. Eventually, 7858 Chinese older adults were included in the final sample for data analysis. Because sampling weights were not released in the sixth wave, we obtained the individual cross-sectional sampling weights from the fifth wave of CFPS to account for the sampling design and improve the estimation precision.

### 3.2. Measurements

Mental health. The eight-item Center of Epidemiological Studies Depression Scale (CES-D) was employed to assess mental health. As a brief version of the 20-item CES-D scale [48], the CES-D 8 is an internationally validated assessment instrument for depressive symptomatology in older populations [49]. The respondents were asked how often during the past week they experienced the following six negative and two positive behaviors or feelings: (1) “I felt depressed”; (2) “I felt everything I did was an effort”; (3) “My sleep was restless”; (4) “I were happy”; (5) “I felt lonely”; (6) “I enjoyed life”; (7) “I felt sad”; (8) “I could not get going”. The response categories for each item were coded as 0 = “Never (less than one day)”, 1 = “Sometimes (1–2 days)”, 2 = “Often (3–4 days)”, and 3 = “Most of the time (5–7 days)”. After reverse-coding the positive items, the sum of scores ranged from 0 to 24. The respondents with a total score ≥ 9 could be identified as having clinically severe depressive symptoms [50,51]. The internal reliability of the eight items in our sample was satisfactory (Cronbach’s alpha = 0.784).

Diet. This study included the environmental, nutritional, and social aspects of diet. The environmental element was measured with the degree of clean cooking, which consisted of energy and water consumption [9]. Cooking fuels were categorized into three broad types: solid fuels (e.g., firewood/straw, coal, and others), gaseous fuels (gas/liquid, natural gas/pipe-line gas, and solar energy/methane), and electricity. The use of water for cooking was measured with the question of “What kind of water does your family normally use for cooking?” Response categories included the following types of water: (1) river/lake water; (2) well/spring water; (3) tap water; (4) mineral/purified/filtered water; (5) rainwater; (6) cellar water; (7) pond water; and (8) other. Among them, tap water could be used in both urban and rural areas and was considered clean water. Hence, a dummy variable of tap water was created (tap water versus non-tap water). Regarding nutrition, the respondents were asked whether they had consumed any meat, which pertained to livestock and marine products, and fresh vegetables/fruits during the past week. Given that almost all respondents had eaten vegetables or fruits (98.15%), the investigation of vegetable/fruit intake may be less meaningful due to the low cell frequency. Amid the COVID-19 pandemic, there was a shortage of meat and food supply, particularly in cities, implying that meat consumption may substantiate older adults’ quality of life. Hence, protein intake was the sole indicator for nutrition (no-meat versus meat-eating). The social aspect of diet was measured with the frequency of dining with family members during the last week, which was a continuous variable, ranging from 0 to 7 nights.

Demographic characteristics. Participants’ age, sex (female versus male), education (in years), marital status (married or cohabited versus unpartnered if being never married, divorced, or widowed) and labor force participation (not working versus working) were included.

Physical health. Physical health was measured using self-rated health, ranging from 1 = “not healthy” to 5 = “very healthy”, and chronic condition. The survey asked the respondents whether they had any doctor-diagnosed chronic disease during the past six months (no versus yes). 

Healthy lifestyle. This study included three key health behaviors: tobacco smoking, alcohol consumption, and physical activity. Tobacco smoking was assessed by asking whether the respondents smoked cigarettes in the past month. Alcohol consumption was a binary variable, with 1 indicating that the respondents drank alcohol three or more times during the past month and 0 otherwise. Physical activity was measured by the frequency of physical exercise during the past year. The responses were rated on a seven-point Likert-type scale from 1 = “less than once per month” to 7 = “at least twice per day”; those who never undertook physical exercise were recoded as 0. 

Family context. We introduced a range of household-level factors, including the area of residence, household size, family income, as well as geographic location. Area of residence was a dummy variable, with 1 = “urban area” and 0 = “rural area”. The residential area was classified based on the level of infrastructure development. Household size and economy were recorded in the fifth wave. Family income was grouped into the following four quartiles: lowest 25%, lower 25%, upper 25%, and highest 25%, using net family income per capita. As for geographic location, the provinces were categorized into four groups, including east, central, west, and northeast, to indicate China’s heterogeneous economic regions.

### 3.3. Data Analysis

Descriptive statistics were presented to profile the respondents of this study. Rural–urban differences in key variables were tested using the Wilcoxon rank-sum test for continuous variables and Pearson’s χ^2^ test for categorical variables. Bivariate correlations between mental health and diet characteristics were calculated using the Spearman correlation. Because depressive symptoms were a count variable and had excessive zeros (10.73%), generalized linear models may be methodically appropriate. We compared the goodness-of-fit indices from linear regression using ordinary least squares (OLS), Poisson, negative binomial regression (NB), zero-inflated Poisson regression (ZIP), and zero-inflated negative binomial regression (ZINB). The ZINB models demonstrated the lowest values of log-likelihood, the Akaike information criterion (AIC) and the Bayesian information criterion (BIC). Previous research has shown that the ZINB models outperformed other generalized linear models in assessing risk factors for depressive symptoms [52]. The ZINB models estimated mental health with the following two models: (a) a logit model for the excessive zeros, that is, those who had the best mental well-being with no depressive symptoms (zero-inflation model: y = 0), and (b) a negative binomial count model, which examined the over-dispersed count of depressive symptoms (high-risk model: y ≥ 1). The high-risk model could estimate the severity of mental distress, which facilitated the clinical interpretation of the roles of dietary factors.

The incidence rate of depressive symptoms was assessed using crude and fully adjusted ZINB models. We first introduced only diet characteristics to explore the unadjusted effects of diet. Then demographic characteristics, physical health, healthy lifestyle, and family context were included as covariates. The mean value of variance inflation factors was 1.41, indicating that no significant issue of multicollinearity was detected in the fully adjusted model. Disparities between rural and urban areas in the associations between diet and mental health were assessed in stratified samples. Supplementary analysis was conducted to evaluate the influence of physical health measures on our results and examine the prevalence of depression. All regression models were weighted using individual sampling weights.

## 4. Results

### 4.1. Descriptive Analysis

The descriptive characteristics of the respondents are presented in Table 1. The mean age of the participants was 60.76 years (standard deviation [SD] = 8.04). A total of 50.9% of the respondents were males. The respondents’ average education was 6.96 years (SD = 4.38). Most were married or cohabited, and more than two-thirds were working. The mean self-rated health was 2.80 (SD = 1.25), and less than a quarter of the participants had chronic diseases during the past half year. In terms of healthy lifestyle, 27.4% of the respondents were smokers, 17.7% were alcohol consumers, and infrequent physical exercise was recorded (mean = 1.78, SD = 2.63). Regarding family context, 56.5% were urban residents, and the average household size was 3.83 persons (SD = 1.88). 

The mean score of depressive symptoms among Chinese older adults was 5.44 (SD = 4.35), suggesting that the overall sample had few mental disorders. However, rural residents had a higher score of depressive symptoms than their urban counterparts (Mean_rural_ = 6.03 versus Mean_urban_ = 4.99). A Wilcoxon rank-sum test showed that the rural–urban difference in mental health was statistically significant (*z* = 11.56, *p* < 0.001).

Regarding diet characteristics, most participants consumed meat over the last week (82.6%). The proportion of rural meat-eaters was significantly lower than that of urban meat-eaters (χ^2^ = 111.74, *p* < 0.001). The mean frequency of family dining together was 6.13 (SD = 2.04) in the total sample. Rural residents had a higher frequency of family dining together than urban residents (Wilcoxon rank-sum test: *z* = 3.04, *p* < 0.01). In terms of cooking energy, 48.9% of the participants used gaseous fuels, 24.7% used electricity, and the remaining 26.5% relied on solid fuels. The use of solid fuels was more common in rural areas (42.8%), whereas urban residents were more likely to use gaseous fuels (62.4%). The difference in the use of cooking fuels was statistically significant (χ^2^ = 1,300, *p* < 0.001). The prevalence of using tap water for cooking was 73.6% among the participants. Nonetheless, more than one-third of rural residents still depended on other less-clean water sources, compared to only about one-fifth in urban areas (χ^2^ = 334.69, *p* < 0.001). Overall, the differences in diet characteristics reflected the regional disparities in the development of public and residential infrastructure and the livelihood situations of Chinese older adults.

Diet characteristics showed significant correlations with depressive symptoms. Protein intake was negatively correlated with depressive symptoms (*r* = −0.12, *p* < 0.001). The correlations between depressive symptoms and the frequency of family dining together (*r* = −0.06, *p* < 0.001), and tap water use (*r* = −0.08, *p* < 0.001) were negative, despite the weak strengths. Using gaseous fuels was negatively correlated with depressive symptoms, while the reliance on solid fuels (*r* = 1.33, *p* < 0.001) was positively associated with depressive symptoms. The correlation between the use of electricity and depressive symptoms was positive but relatively weak (*r* = –0.03, *p* < 0.05). The correlation coefficients between all variables are plotted in Figure 2.

### 4.2. Regression Analysis

Table 2 shows the unadjusted and fully adjusted coefficients of regression models for mental health. The high-risk columns present the point estimation for the severity of depression, while the zero-inflation columns are for zero counts. Regarding the zero-inflation logit models, only the use of gaseous fuels in the crude model was significantly associated with the absence of any depressive symptoms (incidence-rate ratio [IRR] = 1.83, *p* < 0.05). The log odds of having no depressive symptoms were 1.83 times higher among the respondents who used gaseous fuels for cooking than those relying on solid fuels. Hence, we turn to the high-risk sections that fit the count of depressive symptoms.

In the crude model, protein intake was associated with a lower incidence rate of depressive symptoms (IRR = 0.84, *p* < 0.001). The frequency of family dining together was also associated with fewer depressive symptoms (IRR = 0.97, *p* < 0.001). Among cooking fuels, the coefficient of gaseous fuels was statistically significant (IRR = 0.85, *p* < 0.001). Using tap water was associated with a lower incidence rate of depressive symptoms (IRR = 0.88, *p* < 0.001).

After adjusting for demographic characteristics, physical health, healthy behaviors, and family context, the coefficients of protein intake (IRR = 0.89, *p* < 0.001), frequency of family dining together (IRR = 0.98, *p* < 0.001), and tap water (IRR = 0.92, *p* < 0.01) remained statistically significant in the high-risk model. Nonetheless, the effect of gaseous fuels on the count of depressive symptoms was totally attenuated.

We further examined the regional disparities in the associations between diet and mental health, as shown in Table 3. Among rural older adults, only the frequency of dining with family members (IRR = 0.97, *p* < 0.001) and the use of tap water (IRR = 0.89, *p* < 0.001) were significantly associated with a lower risk of depression. In the urban sample, both protein intake (IRR = 0.81, *p* < 0.001) and frequency of family dining together (IRR = 0.98, *p* < 0.05) were associated with a lower incidence rate of depressive symptoms. In addition, the zero-inflation logit models show that none of the diet characteristics was associated with the presence or absence of any depressive symptoms.

### 4.3. Supplementary Analysis

We performed supplementary analysis to check the robustness of our results. First, because concern may exist about the self-reports of physical health, we checked whether different physical health measures could affect the results. We created a dummy variable of physical disability (no versus yes) based on the respondents’ difficulties in carrying out the instrumental activities of daily living. After (a) replacing self-rated health with physical disability or (b) further adjusting for physical disability, the results changed negligibly (not presented but available upon request). Nonetheless, the effect sizes of protein intake decreased slightly, indicating that physical functioning may influence older adults’ food consumption.

Second, we re-assessed the risk of developing clinically severe depression by applying the CES-D 8 cut-off value of 9 [50,51]. The CES-D total score was dichotomized into a dummy variable of depression (1 = depressed if having a total score ≥ 9; 0 = non-depressed if having a total score < 9). Our results showed that the prevalence of being depressed among the older adults was 22.93%, with rural older adults (27.07%) being at a higher risk than their urban counterparts (19.75%). Compared to ZINB models in Table 2 and Table 3, the results of logistic models predicting the risk of being depressed yielded similar results (Appendix A
Figure A1). The frequency of family dining together (odds ratio [OR] = 0.92, *p* < 0.001) and the use of tap water for cooking (OR = 0.67, *p* < 0.001) was significantly related to the lower risk of being depressed among rural residents, whereas only protein intake (OR = 0.57, *p* < 0.001) was significantly associated with urban older adults’ lower risk of being depressed. Although different coding strategies of depressive symptoms showed generally consistent findings, the logistic models revealed the substantive relevance of dietary factors to mental health problems in later adulthood.

## 5. Discussion

To our knowledge, this is the first population-based study to assess the role of multifaceted diet characteristics in shaping healthy aging among Chinese older adults. Using a nationally representative sample, the present study revealed that although older adults had a low risk of mental problems, rural older adults were more disadvantaged. Meanwhile, differences existed in dietary characteristics between rural and urban residents. Regression analyses using the zero-inflated binomial models found that environmental, nutritional, and social aspects of diet were associated with depressive symptoms. In addition, the mental health effects of diet were stratified by the area of residence in China. Our findings contributed to the literature on diet and healthy ageing by fleshing out the understanding of the health effects of various dietary features. Given the ongoing pandemic, the insights from our study could be promising for future aging policies and services by improving the quality of diet and eating among community-dwelling older adults.

Our results showed that Chinese older adults generally had a low risk of clinical psychological distress during the pandemic, as indicated by a mean score of 5.44 and a prevalence rate of 22.93%. It is consistent with extensive evidence that older adults are resilient and can adaptively cope with distressing events [7]. Nevertheless, the results documented a significant rural–urban divide in depressive symptoms among Chinese older adults, with rural participants being more depressed than their urban counterparts (prevalence rates of being depressed: 27.07% versus 19.75%). Because economic inequality is linked with mental health [53], rural Chinese older adults who chronically suffer from material deprivation may be more vulnerable to mental problems amid the pandemic.

The results revealed striking regional differences in diet characteristics. First, clean cooking environments in terms of gaseous fuels and tap water were more commonly used in urban households compared with rural households. Pertaining to protein intake, the proportion of rural meat eaters was significantly smaller than that of urban meat eaters. In contrast, rural residents had a higher frequency of family dining together than their urban counterparts. All these results are consistent with extant evidence [21,42]. The diet and eating patterns may reflect the profound influences of social and economic changes on everyday life among Chinese older adults in different areas. Uneven levels of socioeconomic development between rural and urban areas may lead to differences in the availability of and access to resources and services. It implies that clean cooking resources and food that supports a health-conscious diet may be relatively unaffordable for many older rural families. This could be attributable to individual consumption capabilities and the underinvestment in rural infrastructure. Regarding eating behaviors, having meals alone tends to be slightly more prevalent among urban residents, in line with previous research [35]. This may be partly explained by the rise of individualism in urbanized regions, where people are more likely to maintain independence and autonomy in later life. Nonetheless, the generally high frequency of collective eating suggests that eating together prevails among Chinese older adults. This may be because the pandemic management measures, such as social distancing and home quarantine, have restricted mobility and increased the time spent with family members.

Regression results confirmed that various diet characteristics were associated with depressive symptoms among Chinese older adults. Regarding clean cooking environments, our study showed that using clean water for cooking was a protective factor for mental health, supporting the household environment hypothesis. Clean water is essential to safeguarding mental health by mitigating daily worries and stress, especially for people in poverty [9]. Moreover, in line with the nutrition hypothesis, our results demonstrated that protein intake was associated with a lower incidence rate of depressive symptoms among older adults. This health effect of protein intake is consistent with existing evidence [34]. It suggests that older adults could benefit from weekly protein consumption of meat, milk, or eggs because many Chinese older adults tend to endorse a low-fat diet. Furthermore, the commensality hypothesis is supported by evidence that the frequency of family dining together was associated with better mental well-being among Chinese older adults. As suggested by the notion of commensality, family mealtime should be considered an important opportunity for socialization [35]. Prior research demonstrated that the frequency of eating together is significantly and positively associated with family communication and family satisfaction [54]. The COVID-19 pandemic should also be taken into consideration in the interpretation of results. This is because family dining may contribute to alleviating loneliness and facilitating psychosocial adjustment under stressful circumstances.

This study further showed that the associations between diet and mental health were conditioned by the area of residence. Among rural older adults, tap water, and frequency of family dining together were significantly associated with a lower incidence rate of depressive symptoms. However, regarding urban older adults, protein intake, and frequency of family dining together were related to the lower risk of depression. In terms of protein intake, the emergence of a food shortage due to the sudden outbreak of the COVID-19 pandemic affected peoples’ daily dietary structure [12]. The opportunity for meat consumption in urban districts is highly dependent on food supply, making these regions more prone to food insecurity. In this study, there was a high reliance on meat consumption among urban older adults. Therefore, protein intake may be more likely to affect the mental health of older adults from urban areas. In rural areas, however, engagement in farming and poultry breeding may render older adults’ food access more independent. In addition, the use of tap water was only significantly associated with depressive symptoms among rural older adults. This may because tap water infrastructure that sustains the supply of clean water is well-developed in urban areas but not in many rural villages. Having access to tap water could ensure water security and reduce older residents’ health-related worries in rural areas [55]. Despite differences in nutritional and environmental aspects of diet, the frequency of family dining together affected both urban and rural participants’ depression. More scholarly attention could be paid to those who lack the chance to eat together with family members, such as solitary-living older adults.

By incorporating multidimensional aspects of diet, this study confirmed the protective roles of protein intake, frequency of family dining together, and clean cooking in protecting mental well-being in later years. It makes three key contributions to the literature on diet and healthy aging. First, a theory-guided framework was proposed by bringing together contemporary research on cooking and eating to integrate the nutritional, environmental, and social dimensions of diet. Second, this study deepened the understanding of diet and later-life mental health in the context of the pandemic when food insecurity and social isolation become alarming health issues. Third, by exploring the differential mental health effects of diet between rural and urban older adults, this study raised scholarly awareness of diet-related health disparities.

The results may have significant policy and service implications for promoting healthy aging in China and beyond. Specifically, a policy priority should be placed on improving food supply chains, especially for urban older adults, in a time of unprecedented disruption due to the pandemic. Further investment is required to construct public and residential infrastructure so as to scale up the adoption of clean cooking, particularly clean water, in rural China and in low- and middle-income countries [9]. The food retail industry should improve older people’s accessibility and affordability of functional food that promotes health [56]. Nutritional interventions and home-based social services should recognize the importance of enhancing diet quality and encouraging mealtime interactions among community-dwelling older adults. At the same time, community dining service providers could promote the engagement of family members to address older adults’ social needs. Moreover, tailored psychological interventions are also necessary for screening and alleviating mental problems among underprivileged older adults. Overall, the study results suggest that a healthy diet must be included in future policy agendas, which can be achieved by joint efforts from different professional disciplines.

However, the study limitations should be acknowledged. First, the present study did not include pandemic-related variables because such data are restricted. More research is required to investigate whether and how the pandemic affected diet and mental health among older adults. Second, we relied on brief measures of dietary characteristics. These instruments may render data collection efficient but may lead to measurement errors. We recommend that future research uses relevant validated dietary indicators, such as food-related knowledge, beliefs, and preferences [57,58,59], and include interviewers’ observational information in diet screening. Third, the potential mechanisms that link diet and mental health were not explored in this study. Future research is needed to delve deeper into the biological and psychosocial processes underlying dietary behaviors, such as psychological adjustment to water–food–energy insecurity. Further investigation of the potential interplay between different dietary characteristics affecting older adults’ psychological well-being could cast new insights on dietary pathways to healthy aging. Lastly, in addition to social, nutritional, and physical characteristics, diet bears culturally specific meanings in a Chinese context, as indicated by the old saying “*min yi shi wei tian* (bread is the staff of life)”. For example, the experience of the Great Famine during 1959–1961 has made many Chinese older adults exceptionally appreciative of the food they consume. Despite being beyond the scope of the present study, additional research on the sociocultural meaning of diet and eating in older adulthood is warranted.

## 6. Conclusions

The study investigated the relationships between diet and mental health among Chinese older adults during the COVID-19 pandemic. The findings indicated that desirable environmental, nutritional, and social dimensions of diet, represented by clean cooking, protein intake, and family dining experience, were significantly associated with fewer depressive symptoms among Chinese older adults. In addition, such associations varied across urban and rural areas. Our findings highlight the urgency of promoting clean cooking environments, ensuring equal access to food that supports a health-conscious diet, encouraging family mealtime interactions within the household, and bridging the rural–urban divide in the extra-domestic contexts to achieve healthy aging in the post-pandemic era.

## Figures and Tables

**Figure 1 ijerph-20-01672-f001:**
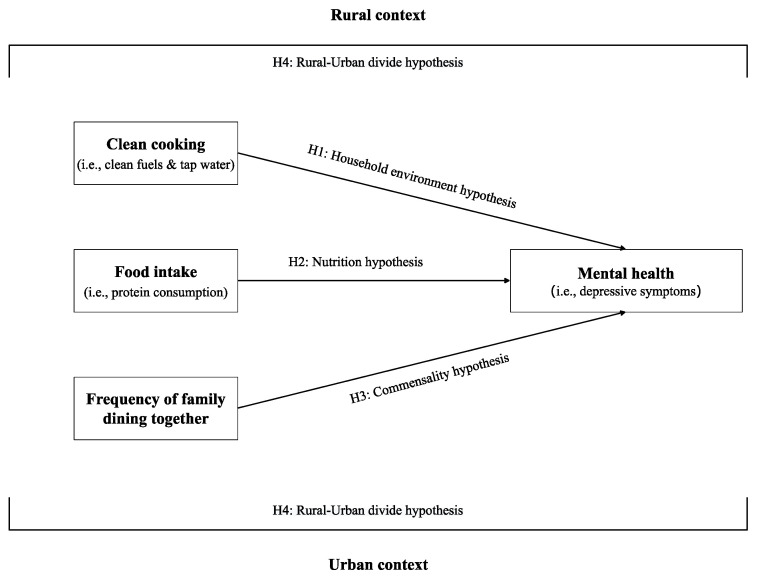
An integrative conceptual framework of diet and mental health in later life.

**Figure 2 ijerph-20-01672-f002:**
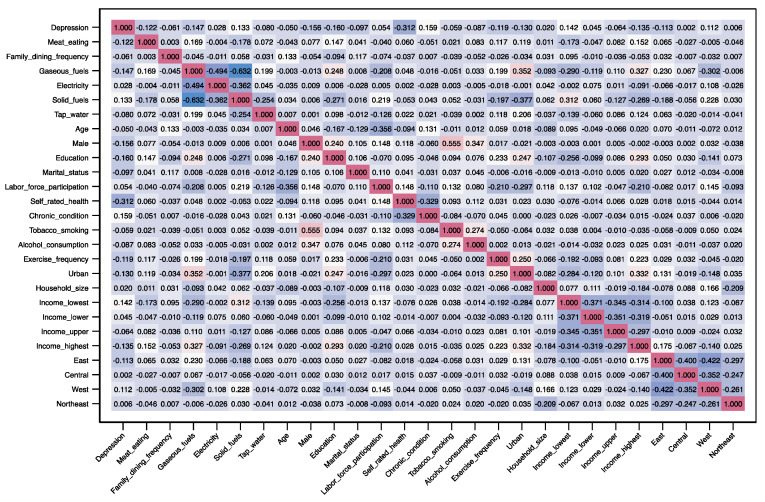
Visualization of Spearman correlation coefficients between all variables. Red colors indicate positive correlations and blue colors denote negative correlations.

**Table 1 ijerph-20-01672-t001:** Characteristics of the sample.

Group	Variable	Total Sample (*N* = 7858)		Rural Residents (*N* = 4088)		Urban Residents (*N* = 3770)	
		Mean/%	SD	Mean/%	SD	Mean/%	SD
Mental health	Depressive symptoms	5.44	4.35	6.03	4.56	4.99	4.13
Diet characteristics	Protein intake						
	No-meat	17.4%		22.4%		13.6%	
	Meat-eating	82.6%		77.6%		86.4%	
	Frequency of family dining together	6.13	2.04	6.19	2.04	6.08	2.03
	Cooking fuels						
	Gaseous fuels	48.9%		31.3%		62.4%	
	Electricity	24.7%		26.0%		23.6%	
	Solid fuels	26.5%		42.8%		13.9%	
	Cooking water						
	Non-tap water	26.4%		35.7%		19.3%	
	Tap water	73.6%		64.3%		80.7%	
Demographics	Age, years	60.76	8.04	61.00	7.88	60.57	8.15
	Sex						
	Female	49.1%		48.6%		49.5%	
	Male	50.9%		51.4%		50.5%	
	Education, years	6.96	4.38	5.80	4.11	7.85	4.38
	Marital status						
	Unpartnered	8.37%		7.72%		8.87%	
	Married or cohabited	91.6%		92.3%		91.1%	
	Labor force participation						
	Not working	32.8%		20.1%		42.5%	
	Working	67.2%		79.9%		57.5%	
Physical health	Self-rated health	2.80	1.25	2.76	1.31	2.83	1.21
	Chronic condition						
	No	77.6%		77.0%		78.1%	
	Yes	22.4%		23.0%		21.9%	
Healthy lifestyle	Tobacco smoking						
	No	72.6%		69.7%		74.9%	
	Yes	27.4%		30.3%		25.1%	
	Alcohol consumption						
	No	82.3%		82.2%		82.5%	
	Yes	17.7%		17.8%		17.5%	
	Exercise frequency	1.78	2.63	1.09	2.21	2.31	2.79
Family context	Area of residence						
	Rural	43.5%		-		-	
	Urban	56.5%		-		-	
	Household size	3.83	1.88	3.96	2.00	3.72	1.78
	Net family income per capita						
	Lowest 25%	24.0%		37.7%		13.4%	
	Lower 25%	26.0%		32.2%		21.2%	
	Upper 25%	26.6%		20.9%		31.0%	
	Highest 25%	23.5%		9.22%		34.4%	
	Geographic location						
	East	36.9%		37.6%		36.3%	
	Central	26.4%		26.9%		26.1%	
	West	22.7%		25.2%		20.7%	
	Northeast	14.0%		10.3%		16.9%	

Data are weighted means or proportions.

**Table 2 ijerph-20-01672-t002:** Zero-inflated negative binomial models of mental health (*N* = 7858).

Group	Variable	Crude Model		Adjusted Model	
		High-Risk	Zero-Inflation	High-Risk	Zero-Inflation
Diet characteristics	Protein intake				
	No-meat (ref.)				
	Meat-eating	0.84 ***	0.93	0.89 ***	0.82
		(0.02)	(0.20)	(0.03)	(0.18)
	Frequency of family dining together	0.97 ***	1.07	0.98 ***	1.05
		(0.01)	(0.06)	(0.01)	(0.05)
	Cooking fuels				
	Gaseous fuels	0.85 ***	1.83 *	0.98	1.39
		(0.03)	(0.51)	(0.04)	(0.43)
	Electricity	0.99	1.52	1.04	1.32
		(0.03)	(0.41)	(0.03)	(0.36)
	Solid fuels (ref.)				
	Cooking water				
	Non-tap water (ref.)				
	Tap water	0.88 ***	1.24	0.92 **	1.16
		(0.03)	(0.27)	(0.03)	(0.24)
Demographics	Age			0.99 ***	1.01
				(0.00)	(0.01)
	Sex				
	Female (ref.)				
	Male			0.88 ***	1.79 **
				(0.03)	(0.36)
	Education			0.99 *	1.02
				(0.00)	(0.02)
	Marital status				
	Unpartnered (ref.)				
	Married or cohabited			0.85 ***	1.46
				(0.03)	(0.54)
	Labor force participation				
	Not working (ref.)				
	Working			1.05	0.64 *
				(0.03)	(0.12)
Physical health	Self-rated health			0.86 ***	1.55 ***
				(0.01)	(0.11)
	Chronic condition				
	No (ref.)				
	Yes			1.09 **	0.62 *
				(0.03)	(0.14)
Healthy lifestyle	Tobacco smoking				
	No (ref.)				
	Yes			1.05	1.04
				(0.03)	(0.20)
	Alcohol consumption				
	No (ref.)				
	Yes			0.97	1.08
				(0.03)	(0.23)
	Exercise frequency			0.99	1.08 *
				(0.00)	(0.04)
Family context	Area of residence				
	Rural (ref.)				
	Urban			0.95	0.93
				(0.04)	(0.19)
	Household size			0.99	1.04
				(0.01)	(0.05)
	Net family income per capita				
	Lowest 25% (ref.)				
	Lower 25%			0.96	1.34
				(0.03)	(0.29)
	Upper 25%			0.88 ***	1.07
				(0.03)	(0.28)
	Highest 25%			0.82 ***	1.18
				(0.04)	(0.34)
	Geographic location				
	East			0.94	1.14
				(0.04)	(0.28)
	Central (ref.)				
	West			1.04	0.86
				(0.04)	(0.27)
	Northeast			0.96	0.82
				(0.04)	(0.29)

Incidence-rate ratios are reported (exponentiated coefficients); standard errors are in parentheses; the data are weighted. * *p* < 0.05, ** *p* < 0.01, *** *p* < 0.001.

**Table 3 ijerph-20-01672-t003:** Zero-inflated negative binomial models of mental health stratified by area of residence.

Diet Characteristics	Rural Residents (*N* = 4088)		Urban Residents (*N* = 3770)	
	High-Risk	Zero-Inflation	High-Risk	Zero-Inflation
Protein intake				
No-meat (ref.)				
Meat-eating	0.95	0.81	0.81 ***	0.77
	(0.04)	(0.25)	(0.03)	(0.33)
Frequency of family dining together	0.97 ***	1.04	0.98 *	1.04
	(0.01)	(0.05)	(0.01)	(0.09)
Cooking fuels				
Gaseous fuels	0.96	1.50	1.04	2.76
	(0.04)	(0.54)	(0.07)	(2.81)
Electricity	1.04	1.14	1.09	3.19
	(0.04)	(0.35)	(0.07)	(3.40)
Solid fuels (ref.)				
Cooking water				
Non-tap water (ref.)				
Tap water	0.89 ***	0.92	0.96	1.53
	(0.03)	(0.25)	(0.04)	(0.57)

Incidence-rate ratios are reported (exponentiated coefficients); standard errors are in parentheses; the data are weighted; all models are adjusted for demographics (age, sex, education, marital status, and labor force participation), physical health (self-rated health and chronic condition), healthy lifestyle (tobacco smoking, alcohol consumption, and exercise frequency), and family context (household size, net family income per capita, and geographic location). * *p* < 0.05, *** *p* < 0.001.

## Data Availability

The data from the China Family Panel Studies can be accessed at https://www.isss.pku.edu.cn/cfps/en/data/public/index.htm (accessed on 2 January 2023).

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
