# Peer review of "Healthy Aging at Family Mealtimes: Associations of Clean Cooking, Protein Intake, and Dining Together with Mental Health of Chinese Older Adults amid COVID-19 Pandemic"

_ijerph, 2023, doi:10.3390/ijerph20031672_

Round 1

Reviewer 1 Report

The manuscript entitled „Healthy Aging During Family Mealtime: The Importance of Clean Cooking, Protein Intake, and Dining Together” is an interesting and well-presented overview of some dietary aspects related to psychological elements of healthy ageing. The sample and the used statistical analysis are also appropriate.

I identified some minor issues during my review, most of them connected to the adequate description and interpretation of the results. My recommendations are the following:

-        I suggest adding “in China” to the end of title of the manuscript

-        It should be emphasized in all sections of the manuscript that the study examines mental health only from dietary perspective (within social dimensions), so other possible effects (e.g. childhood experiences, family background, previous traumas) are not presented.

-        All the factors of the presented framework (Fig. 1) are measured through a low number of variables (e.g. ‘clean cooking’ through ‘fuel types’ and ‘clean water’ variables) only. It would be useful to provide some recommendations for further studies during the limitation part of the discussion to extend questionnaires with additional relevant variables for more detailed results.

-        Please mention the possible correlations between the variables. For example, the income plays a significant role on mental health according to the results. This cannot be strictly separated from other effects (e.g. it can be assumed that people living in the rural area have a lower income).

-        Recommended modifications to the Fig. 1:
- More precise to use ‘i.e.’ except ‘e.g.’ in this case
- Use ‘frequency of family dining together’ instead of ‘Family dining experience’
- The stickman is not necessary on the figure
- What do the arrows from 'clean cooking' and 'family dining experience' factors to 'food intake' factor represent? The results of the article confirm that all three factors affect 'mental health,' but the relationship between the factors is not mentioned. Based on the figure, we can conclude that these 2 factors affect the mental factors indirectly as well. If there is an explanation for this, then it would be necessary to complete the manuscript with this description. If there is not, then I recommend deleting the arrows between these factors.

-        It is a strength of the paper that the scope of the article is well-defined during the introduction, but I suggest to consider including some references on related consumer research dealing with health-related dietary choices of older adults:

Febian, F. I., Annuar, S. N. S., & Memon, M. A. (2021). Functional food consumption among older consumers in Malaysia: A Health Belief Model perspective. British Food Journal, 123(8), 2880–2892.

Liu, R., & Grunert, K. G. (2020). Satisfaction with food-related life and beliefs about food health, safety, freshness and taste among the elderly in China: A segmentation analysis. Food Quality and Preference, 79, 103775.

Szakos, D., Ózsvári, L., & Kasza, G. (2020). Perception of older adults about health-related functionality of foods compared with other age groups. Sustainability, 12(7), 2748.

van der Zanden, L. D., van Kleef, E., de Wijk, R. A., & van Trijp, H. C. (2014). Knowledge, perceptions and preferences of elderly regarding protein-enriched functional food. Appetite, 80, 16-22.

-        Line 309: ’foods that support health-conscious diet’ instead of ’healthy foods’

-        Data availability statement: the link does not work

Reviewer 2 Report

ID: ijerph-2111226

Title: Healthy Aging During Family Mealtime: The Importance of Clean Cooking, Protein Intake, and Dining Together

Thank you for providing a chance to review this manuscript.

Comment: major revision.

Detailed information:

Title

I think your title could be properly polished. Highlight the time factor and area -- during the COVID-19 pandemic and in China.

Abstract

        1) A little bit confusing in my opinion. Dividing the paragraph into “Objective”, “Method”, “Results”, and “Conclusion” four parts is a great way to make the meaning clearer. 2) Simplify the text in the abstract appropriately. 3) Please list specific results (values).

1. Introduction

Line 45-51, page 2: Previous studies have only mentioned that eating alone increases the risk of depression in older adults. Have other studies mentioned Clean Cooking, Protein Intake? Is that the innovation of your research? And, please explain in detail some terms such as clean cooking.

Line 53-62, page 2: Part of the sentence makes me confused. It will make readers more interested to express your research purpose and content clearly.

2. Literature review

2.1 Multifaceted diet and healthy aging: The content is too long, please simplify it.

2.3 The present study

Line 133-145, page 3: There is little need for this section to exist, and it is heavily repeated in the last paragraph of the introduction. It is suggested to simplify or delete directly.

3. Materials and Methods (Please note that the number before the heading is correct !!!)

3.1 Data

Line 175-182, page 5: What is the inclusion criteria? How to carry out quality control?

3.2 Measurements

    Physical health: Is it unscientific to use self-rated health as a measure?

4. Results

4.1. Descriptive results

Line 271, page 6: It’s 60.76.

4.2. Regression results: Is there a collinearity problem in the regression model and how to solve it?

5. Discussion

        In my opinion, your discussion section is well written and logically structured. However, the biggest problem of your article is that according to your research results, the score of CESD-8 of the elderly in China is about 5.4, indicating that there is almost no mental disorder. This is inconsistent with the previous research results, and further discussion is necessary.

Your article proposes a new dietary framework that includes environmental, nutritional, and social aspects that other research could use and learn from. However, some paragraphs still have the problem of structure confusion, especially the preface. Then, remove or simplify some unnecessary words and statements to make the article look cleaner. Furthermore, pay attention to the normalization of the serial number. Attention to detail is icing on the cake!!

Thank you and my best,

Your reviewer

Reviewer 3 Report

1. What is the main contribution of this study? 

2. What is the main impact of the literature review on this study? 

Round 2

Reviewer 2 Report

ID: ijerph-2111226

Title: Healthy Aging During Family Mealtime: The Associations of Clean Cooking, Protein Intake, and Dining Together with Mental Health of Chinese Older Adults Amid COVID-19 Pandemic

Thank you for providing a chance to review this manuscript.

Comment: minor revision.

Detailed information:

1.Introduction and 2. Literature Review

In the background and literature review, please describe it in the order of “Clean Cooking”, “Protein Intake” and “Dining Together”.

3. Materials and Methods

3.2 Measurements

Line 269, page 4: How reliable and valid the scale has been in the past? What is the cut-off value for depression on this scale?

Thank you and my best,

Your reviewer

Author Response

1.Introduction and 2. Literature Review

In the background and literature review, please describe it in the order of “Clean Cooking”, “Protein Intake” and “Dining Together”.

Authors’ response: Thanks for pointing out this issue. We have revised the organization of background and literature review.

In the background, the revised text is in Lines 38-50, pages 1-2:

“Notably, several strands of research have explored the roles of cooking and eating in mental well-being. Environmental research has advocated for the notion of clean cooking and argued that using clean water and energy for cooking is salutary for sustainable livelihoods and human health [9,10]. In addition, nutrition research has indicated that food intake is crucial for maintaining mental health [11]. Amidst the COVID-19 pandemic, a global food crisis emerged due to the shortage of food supply and income shrinkage,which has substantially affected individual eating behaviors [12]. In 2021, the World Health Organization released an action framework for healthy diet that emphasizes the necessity for policymakers to improve the availability and quality of food in public settings, such as community centers [13]. However, diet pertains to what people eat, how they cook, and, more importantly, whom they dine with. Emerging studies from east Asia, for example, showed that eating alone may increase the risk of depression [14] and frailty [15] in old age.”

  1. Materials and Methods 

3.2 Measurements

Line 269, page 4: How reliable and valid the scale has been in the past? What is the cut-off value for depression on this scale?

Authors’ response: The CES-D 8 is a widely used tool for screening depression and has been validated in older people. We have provided information about the validity of the scale in the measures in Lines 166-168 and Lines 175-176, pages 4-5:

  “As a brief version of the 20-item CES-D scale [48], the CES-D 8 is an internationally validated assessment instrument for depressive symptomatology in older populations [49]…… After reverse-coding the positive items, the sum of scores ranged from 0 to 24. The respondents with a total score ≥ 9 could be identified as having clinically severe depressive symptoms [50,51].”

The cut-off values of mental health scales are typically used for clinical purposes.  Briggs et al. (2018) recommended to use 9 as the cut-off score of CES-D 8 scale. It means that participants with a score larger than or equal to 9 may have severe depressive symptoms, and, therefore, need psychological assistance. This cut-off score has been used in studies of Chinese sample. In our study, we examined the binary variable of depression in supplementary analysis to check whether different coding strategies could produce consistent findings and highlight dietary protective factors of depression.

The revised text could be found in Lines 345-348, page 11:

“we re-assessed the risk of developing clinically severe depression by applying the CES-D 8 cut-off value of 9 [50,51]. The CES-D total score was dichotomized into a dummy variable of depression (1 = depressed if having a total score ≥ 9; 0 = non-depressed if having a total score < 9).”

References:

Briggs, R.; Carey, D.; O’Halloran, A.M.; Kenny, R.A.; Kennelly, S.P. Validation of the 8-Item Centre for Epidemiological Studies Depression Scale in a cohort of community-dwelling older people: Data from The Irish Longitudinal Study on Ageing (TILDA). Eur. Geriatr. Med. 2018, 9, 121–126, doi:10.1007/s41999-017-0016-0.

Thank you very much!